Effects of particle size on physicochemical and functional properties of superfine black kidney bean (Phaseolus vulgaris L.) powder

Sun Xianbao 1 2
Zhang Yuwei 3
Li Jing 4
Aslam Nayab 5
Sun Hanju 1 2 6
Zhao Jinlong 1
Wu Zeyu 1 2
http://orcid.org/0000-0002-1155-4928 He Shudong 1 2 6 shudong.he@hfut.edu.cn
1 Engineering Research Center of Bio-process, Ministry of Education, Hefei University of Technology , Hefei , China
2 School of Food and Biological Engineering, Hefei University of Technology , Hefei , China
3 School of Food Science and Engineering, Shaanxi Normal University , Xi’an , China
4 Department of Biological and Environmental Engineering, Hefei University , Hefei , China
5 Institute of Home & Food Sciences, Government College University , Faisalabad , Pakistan
6 Anhui Province Key Laboratory of Functional Compound Seasoning, Anhui Qiangwang Seasoning Food Co., Ltd. , Jieshou , China
Tan Chin Ping
Electronic publication date: 2019 Feb 4
Publication date: 2019
Volume: 7
Electronic Location ID: e6369
Received 2018 Jul 5; Accepted 2018 Dec 29
Copyright: © 2019 Sun et al.
Copyright year: 2019
Copyright holder: Sun et al.
License: This is an open access article distributed under the terms of the Creative Commons Attribution License, which permits unrestricted use, distribution, reproduction and adaptation in any medium and for any purpose provided that it is properly attributed. For attribution, the original author(s), title, publication source (PeerJ) and either DOI or URL of the article must be cited.
License URL: https://creativecommons.org/licenses/by/4.0/

Keywords: Black kidney bean, Superfine grinding, Eccentric vibratory mill, Particle size, Physicochemical properties, Cholesterol adsorption capacity, Antioxidant activities

Funding: National Natural Science Foundation of China 31701524, 31771974 Anhui Provincial Natural Science Foundation 1708085MC70 Fundamental Research Funds for the Central Universities JZ2018HGTB0245 Anhui Provincial Science and Technology Major Project 16030701081, 16030701084 Financial Grant from China Postdoctoral Science Foundation 2017M611208, 2018T110211 This work was supported by the National Natural Science Foundation of China (No. 31701524, No. 31771974), the Anhui Provincial Natural Science Foundation (No. 1708085MC70), the Fundamental Research Funds for the Central Universities (No. JZ2018HGTB0245), the Anhui Provincial Science and Technology Major Project (No. 16030701081, No. 16030701084), and the Financial Grant from China Postdoctoral Science Foundation (No. 2017M611208, No. 2018T110211). No additional external funding was received for this study. The funders had no role in study design, data collection and analysis, decision to publish, or preparation of the manuscript.

==============================
Black kidney bean (Phaseolus vulgaris L.) powder (BKBP) with particle sizes of 250–180, 180–125, 125–75, 75–38, and <38 μm was prepared by using coarse and eccentric vibratory milling, respectively. Physicochemical properties, cholesterol adsorption, and antioxidant activities of powders were investigated. Size and scanning electron microscopy analyses showed that particle size of BKBP could be effectively decreased after the superfine grinding treatment, and the specific surface area was increased. Flow properties, hydration properties, thermal stability, and cholesterol adsorption efficiency significantly improved with the reducing of particle size. The superfine powder with sizes of 75–38 or <38 μm exhibited higher antioxidant activity via 2,2-diphenyl-1-picryhydrazyl, hydroxyl radical-scavenging, and ferrous ion-chelating assays. The results indicated that the BKBP with a size of <38 μm could serve as a better potential biological resource for food additives, and could be applied for the development of low-cholesterol products.

Introduction

As an essential crop, kidney beans (Phaseolus vulgaris L.) are particularly popular in Africa, Latin American, and Asia (Beninger & Hosfield, 2003), and consumed as a human food source throughout the world representing 50% of the grain legumes (Camara, Urrea & Schlegel, 2013). Potential benefits to human health have been explored during the kidney beans consumption, including lowering postprandial glucose and insulin responses, preventing obesity, reducing the risk of cardiovascular diseases and preventing cancers because of the high contents in protein, fiber, vitamin B, and chemically diverse micronutrient compositions (Ai et al., 2016; Anton, Fulcher & Arntfield, 2009; Mitchell et al., 2009). Furthermore, the phenols are rich in kidney beans, which will lead to greater anti-oxidative activity and will be beneficial for preventing oxidative damages (Camara, Urrea & Schlegel, 2013; Lee & Yoon, 2015).

Recently, due to the nutritional and economical values, kidney bean powders have been employed as a food ingredient in the manufacture of value-added products (Anton, Fulcher & Arntfield, 2009; Malav et al., 2016); however, the properties of black kidney bean powder (BKBP) have not been well documented to date. As a necessary process, controlling the particle size in the grinding process is of importance, as it will influence powder behaviors during storage, handling, and processing (Lee & Yoon, 2015). Powder with a larger particle size might be unwieldy for extraction and leaching, and would lengthen heat treatment for blanching and/or cooking (Barbosa-Canovas et al., 2005). In addition, various changes in powder color, texture, and bioactive compounds as well as taste acceptability would be also dependent on variations of particle size (Lee & Yoon, 2013; Liu et al., 2000; Zhu et al., 2010). Thus, searching for appropriate particle size for BKBP would be necessary to improve the application in nutraceuticals and functional food products, as well as a potential and novel biomaterial.

As a useful and novel process technology, superfine grinding has been widely applied in the ceramic, electric material, and chemical powder product developments due to the well contributions to surface effect, mini-size effect, mechanical property, and chemical and/or catalytic properties (Zhao et al., 2009b). Currently, because of the increasing processing requirements, superfine grinding methods, such as ball milling, jet milling, and high-pressure expansion, have begun to be applied in the food industry. Tan et al. (2015) reported that waxy and high-amylose corn starches had a lower viscosity and better pasting stability via planetary ball-milling. Phat et al. (2015) confirmed that Hericium erinaceum powder prepared via jet milling had good water solubility and swelling capacity, which would be suitable to manufacture instant and convenient foods. In the previous study, superfine okra powders exhibited a higher cholesterol adsorption capacity compared with the coarse (Chen et al., 2015b), indicating that the superfine powder might be a potential bio-absorbent for low-cholesterol food developments. It has also been reported that Astragalus membranaceus, wheat (Triticum aestivm L.) bran, and okra (Abelmoschus esculentus) superfine powders have an increased flowability (Chen et al., 2015b; He et al., 2018; Zhao et al., 2010). Several studies found that superfine grinding treatment could enhance the antioxidant activities of some food powders, such as red rice (Oryza sativa L.), Qingke (hull-less barley), and winter wheat (Triticum aestivm L.) bran superfine powders (Chen et al., 2015a; He et al., 2018; Zhu, Du & Xu, 2015) through altering the molecular weight and solution behavior of bioactive components. However, Anguita et al. (2006) reported that a reduction in the particle size of barley after grinding was accompanied by a decrease of water retention capacity (WRC). Choi et al. (2018) and Liu et al. (2015) found that increasing degrees of milling significantly reduced phenolics in rice flours. It has also reported that the antioxidant ability of wheat bran powder was coarse > medium > fine size (Brewer et al., 2014). Then, the reverse phenomena indicated the physicochemical properties seemed to be unpredictable, and would be related to particle size reduction, various grinding treatments, and raw materials. Thus, the effects of superfine grinding treatment on the physicochemical and functional properties of BKBP should be explored due to the little information.

The newly-designed eccentric vibratory mill is remarkable nowadays, exhibiting a decisive intensification of the impact force among the grinding rollers for improved effectivity (Gock & Kurrer, 1999). In addition, the power consumption of eccentric vibratory milling is significantly decreased (up to 50% compared to conventional vibratory tube mills), due to the decrease of the ratio between kerb mass and payload, as well as the rational bearing load (Beenken, Gock & Kurrer, 1996), then it is increasingly used for fine grinding and pulverization of raw materials on an industrial scale (Baláž & Dutková, 2009; Godočíková et al., 2006). Thus, the BKBP was developed via eccentric vibratory milling in this study, and the effects of particle size on physicochemical, microstructural, cholesterol adsorption, and antioxidant properties of the resulting powders were investigated. The results are favorable for the development of value-added products using the BKBP.

Materials and Methods

Materials

Black kidney beans (P. vulgaris L.) were obtained from a local supermarket in Hefei, Anhui Province, China, and with a species authentication by Heilongjiang Crops Variety Examination Committee (Heilongjiang Province, China). Ferrozine, 2,2-diphenyl-1-picryhydrazyl radical (DPPH), ferrous sulfate, salicylic acid, and cholesterol were purchased from Sinopharm Chemical Reagents Co. (Shanghai, China). All other used chemicals were of analytical grade.

Micronization processing of black kidney bean

The dried black kidney beans were milled coarsely by a domestic disc-mill (DS-T200A model, Shanghai Dingshuai Electric Co., Ltd., Shanghai, China) for a 3 min discontinuous grinding, and then screened through 250–180 μm sieves. The resulting coarse samples were re-milled through an eccentric vibratory mill (XDW-6J model; Jinan Micro Machinery Co., Ltd., Shandong, China) for 10 min, and superfine powders with the particle sizes of 180–125, 125–75, 75–38 and <38 μm were then obtained via sieving. The eccentric vibratory mill was consisted of cylindrical-like elastically suspended grinding pipes, and kept the frequency of an unbalanced drive constant at 1,000 rpm during grinding. The circulating cold water was applied to maintain a low temperature.

Particle size distribution and specific surface area analysis

Particle size distribution of BKBP was analyzed via laser diffraction particle size analyzer (Mastersizer 2000; Malvern Instruments Ltd., Worcestershire, UK). The samples were dispersed in the ethanol before measured, and the volume weighted mean diameter of D[4,3], as well as the selected percentile points of D10, D50, and D90, which represent 10%, 50%, and 90% volume of the particle mass diameter that is smaller than the size indicated, respectively, was used to characterize the particle size distribution of the superfine powder. The specific surface area (m2/g) was also calculated based on the volume distribution by the particle size analyzer.

Scanning electron microscopy analysis

Morphological characterization of the BKBP particles was performed using scanning electron microscope (SEM) (JSM-6490LV; JEOL Ltd., Tokyo, Japan) at an operating voltage of 20 kV with working distance of 11 mm.

Color analysis

The color of sample was detected via an automatic color difference meter (WB2000-IXA; Shanghai Exact Science Instrument Ltd., Shanghai, China) using the Hunter scale of L*, a*, and b* values as indicators.

Flow property analysis

The flow properties of BKBP were determined via bulk density (g/mL), tapped density (g/mL), angle of repose (°), and angle of slide (°) using a powder integrative characteristic testing instrument (BT-1000; Bettersize Instruments Ltd., Liaoning, China) (He et al., 2018).

Water holding capacity and water retention capacity analyses

The water holding capacity (WHC) was determined using the method of Zhao et al. (2010). The weights of cleaned centrifuge tubes (M0) and dry BKBP samples (M1) were measured, and the samples were then dispersed in the water with a ratio of 0.05:1 (w/w) and incubated at 60 °C for 10, 20, 30, 40, 50, and 60 min, respectively. After centrifugation for 20 min at 5,000 rpm, the supernatant was removed, and the centrifuge tubes with the powder (M3) were weighed. The WHC of BKBP was calculated as follows:WHC (g/g)=(M3−M0−M1)M1

Water retention capacity was defined as the quantity of water that remains bound to the hydrated fiber following application of an external force. The samples (M3) were dried at 105 °C for 2 h, and then weighed (M4) again to calculate the WRC as follows:WRC (g/g)=(M3−M4)M4

Thermal property analysis

The thermal property was analyzed via the differential scanning calorimetry (DSC) method using a TA ultrasensitive differential scanning microcalorimeter (Model TA Q200; TA Instruments Co., New Castle, DE, USA). Eight milligrams of each sample were put into a hermetic aluminum pan and heated from 20 to 220 °C at a rate of 10 °C/min in a 50 mL/min nitrogen flow, using an empty aluminum pan as reference. Each curve obtained by the instrument was further analyzed via Universal Analysis 2000 software (TA Instruments Co., New Castle, DE, USA).

Cholesterol adsorption capacity analysis

The cholesterol adsorption capacity was expressed as the quality of adsorbed cholesterol for per gram of BKBP, which was estimated by the method of Chen et al. (2015b). The cholesterol solution with different concentrations was prepared in glacial acetic acid. The BKBP was added in cholesterol solution with a selected mass ratio, and then placed in a shaker water bath at 37 °C for 90 min at 90 rpm. At the end of adsorption, two mL of the supernatant were used for cholesterol estimation. The cholesterol adsorption capacity was calculated using the following formula:Cholesterol adsorption capacity (mg/g)=[V(ρ0–ρ)]m

Where V represents the volume of the cholesterol solution, ρ0 and ρ represent the concentrations of cholesterol solution before and after adsorption, respectively, and m represents the weight of BKBP. Effects of the particle size, powder dosage, initial concentration of cholesterol, absorption time, and absorption temperature on cholesterol adsorption capacity were evaluated.

Antioxidant activity analysis

The antioxidant activity was determined via the scavenging activities of DPPH and hydroxyl free radicals. Two milliliters of BKBP solution (five mg/mL) were mixed with 2.5 mL DPPH-ethanol solution (100 μM) for a 30 min reaction at 37 °C. Then, the mixture was centrifuged at 10,000 rpm for 10 min, and the absorbance of the supernatant (Abssample) was recorded at 517 nm. Blank absorbance (Absblank) was measured using methanol to replace the sample. Vitamin C (VC, five mg/mL) was used as positive control. The DPPH radical scavenging activity (%) was calculated using the equation of [(Abssample − Absblank)/Absblank] × 100% (Andrade et al., 2017).

The hydroxyl radical scavenging activity (%) was estimated following a previously reported method (Zhao et al., 2015). Two milliliters of BKBP solution (five mg/mL) was used for testing. The reaction mixture solution was centrifuged at 10,000 rpm for 10 min to determine the absorbance of the supernatant at 510 nm. Methanol was applied to determine the blank absorbance (Absblank), and the hydroxyl radical scavenging activity (%) of BKBP was calculated by [(Abssample − Absblank)/Absblank] × 100%.

In addition, the Fe2+ chelating capacity was also measured. One milliliter of BKBP solution (five mg/mL) was mixed with 2 M FeCl2 solution (0.1 mL) under addition of 0.2 mL of five mM ferrozine and was left standing for 10 min. The supernatant after centrifugation was recorded at 517 nm and the reaction mixture without sample was used as a blank (Absblank); then, the Fe2+ chelating activity (%) was obtained via the equation of [(Abssample − Absblank)/Absblank] × 100% (He et al., 2018).

Statistical analysis

All experiments were repeated and analyzed at least in triplicate. Results were expressed as the mean ± SD, and one-way analysis of variance was employed to determine the significant differences between the means at P < 0.05 using SPSS version 13.0 (SPSS Inc., Chicago, IL, USA).

Results

Particle properties and microphotographs of BKBP

The particle size distribution and specific surface area of BKBP were presented in Table 1. With particle size decreasing, all cumulative undersize centiles (D10, D50, and D90) of BKBP significantly (P < 0.05) decreased. D[4,3] values of the powder decreased from 226.658 to 24.835 μm for a particle size ranging from 250–180 to <38 μm (Table 1). Furthermore, the specific surface area increased with the decrease of particle size, and the BKBP with the smallest particle size (<38 μm) showed the highest specific surface area of 0.520 m2/g, suggesting that the surface parameter of BKBP was negatively related to the projected size of the corresponding particle.

Table 1 Particle size distributions and specific surface areas of the BKBP obtained from the laser diffraction method.

Powder particles (μm)	Equivalent diameter particles accounted for by measuring the proportion (μm)	Specific surface area (m2/g)	
D10	D50	D90	D(4,3)	
250–180	54.366 ± 2.382a	257.167 ± 5.252a	500.742 ± 5.667a	226.658 ± 3.875a	0.125 ± 0.010d	
180–125	24.460 ± 2.029b	217.081 ± 3.820b	362.479 ± 6.836b	214.801 ± 7.033b	0.133 ± 0.009d	
125–75	25.671 ± 0.337b	140.998 ± 3.799c	255.877 ± 2.529c	146.407 ± 5.660c	0.176 ± 0.017c	
75–38	9.694 ± 1.424c	36.594 ± 1.328d	86.295 ± 5.650d	45.962 ± 3.114d	0.297 ± 0.015b	
<38	4.810 ± 0.533d	20.706 ± 2.025e	51.331 ± 4.988e	24.835 ± 5.494e	0.520 ± 0.026a	
Note:

The results were expressed as mean ± standard deviation. Data in the same column with different letters were significantly different (P < 0.05).

The shape and surface morphology of BKBP were observed using SEM (Fig. 1). As the particles size decreased, it was possible to see the transition of typical blocky shape (coarse powder) into short ones (Fig. 1C, 125–75 μm), until very small parts and fragments were achieved (Figs. 1D and 1E, <38 μm). From Figs. 1B–1E, with the improvement of mechanical force, the transformation of BKBP from an ordered structure to a disordered structure was clearly presented via the breakage of intermolecular bonds as well as the reduction of particle size. It was notable that Figs. 1D and 1E exhibited an increased aggregation of BKBP, due to the various shapes of black kidney bean particles resulted from the extensive milling combination of flattening, aggregation and fracture. Under the higher magnification (Figs. 1F–1J), it could be clearly seen that the particles surface tends to be flat and smooth with the size decreasing.

Figure 1 SEM images of BKBP with different particle sizes.

(A) 250–180 μm, (B) 180–125 μm, (C) 125–75 μm, (D) 75–38 μm, (E) <38 μm with scale bar 100 μm; (F) 250–180 μm, (G) 180–125 μm, (H) 125–75 μm, (I) 75–38 μm, and (J) <38 μm with scale bar 10 μm.

Color

As listed in Table 2, L* increased slightly (P < 0.05) when the BKBP size decreased from 250–180 to 75–38 μm, and no significant difference (P > 0.05) in lightness was found between the sample sizes of 75–38 μm and <38 μm. Furthermore, an increase in a* value could be observed, but it was difficult to visually obtain due to the smaller variance. The b* value decreased from 30.654 to 15.805 with BKBP size decreasing from 250–180 to 125–75 μm, while increasing to 35.653 at the particle size <38 μm.

Table 2 Color and flow properties of the BKBP.

Powder particles (μm)	Color	Flow property	
L*	a*	b*	Bulk density (g/mL)	Tapped density (g/mL)	Angle of repose (°)	Angle of slide (°)	
250–180	90.389 ± 0.119d	−4.748 ± 0.127d	30.654 ± 0.872b	0.439 ± 0.022a	1.435 ± 0.525c	51.878 ± 1.102a	45.452 ± 0.833a	
180–125	91.331 ± 0.312c	−3.583 ± 0.008c	19.092 ± 1.028c	0.416 ± 0.018ab	1.684 ± 0.329bc	49.013 ± 0.330b	41.653 ± 0.243b	
125–75	91.598 ± 0.006bc	−3.216 ± 0.052c	15.805 ± 1.150d	0.396 ± 0.009b	1.971 ± 0.070bc	49.968 ± 1.029b	38.280 ± 2.049c	
75–38	91.974 ± 0.228a	−3.550 ± 0.309b	20.722 ± 0.877c	0.369 ± 0.010c	2.214 ± 0.104ab	46.784 ± 0.144c	35.155 ± 0.638d	
<38	91.711 ± 0.015ab	−2.609 ± 0.205a	35.653 ± 0.649a	0.364 ± 0.005c	2.645 ± 0.220a	43.282 ± 0.936d	33.259 ± 1.550d	
Note:

The results were expressed as mean ± standard deviation. Data in the same column with different letters were significantly different (P < 0.05).

Flow property

To evaluate the flowability of BKBP, the integrative characteristics of powder were analyzed. As the particle size decreased from 250–180 to <38 μm, the bulk density decreased from 0.439 to 0.364 g/mL, and the largest bulk density (0.439 g/mL) was found in the particle size of 250–180 μm (Table 2). In contrast, the tapped density of BKBP increased from 1.435 to 2.645 g/mL with BKBP size decreasing from 250–180 to <38 μm. The values of tapped density were significantly higher than the bulk density. Moreover, the angle values of repose and slide decreased significantly (P < 0.05) with the reduction of particle size. The BKBP with a particle size of <38 μm had the lowest angles of repose (43.282°) and slide (33.259°).

Hydration property

The hydration property of BKBP was determined by WHC and WRC assays. With the reduction of particle sizes from 250–180 to <38 μm, the WHC values of BKBP increased, ranging from 5.98 to 6.26 g/g, 6.03 to 6.87 g/g, 6.18 to 7.41 g/g, 6.28 to 7.86 g/g, 6.31 to 7.81 g/g, and 6.44 to 8.03 g/g for 10, 20, 30, 40, 50, and 60 min of soaking (Fig. 2A), respectively. A similar tendency was also found in the WRC assay for the BKBP under the same soaking conditions (Fig. 2B). Thus, the hydration property of the BKBP with a particle size of <38 μm was higher. It was also worth mentioning that the WHC values of different sized BKBP increased slowly during the initial 10 min soaking.

Figure 2 Hydration properties of BKBP with particle sizes of 250–180, 180–125, 125–75, 75–38, and <38 μm for soaking time 10–60 min.

(A) Water holding capacity (g/g) and (B) water-retention capacity (g/g).

Thermal property

The thermal property of BKBP with different sizes was further analyzed via DSC curves (Fig. 3). Compared to the endothermic peaks (Tm) observed in the curve of coarse powder, the peak around 97.07 °C disappeared in the analyses of superfine powder with sizes of 180–125, 125–75, 75–38, and <38 μm. Notably, an intense endothermic peak was found from 128.56 to 178.10 °C in all curves, and the peak temperatures exhibited a significant increasing tendency with the decreasing of particle size.

Figure 3 Average DSC curves of BKBP with particle sizes of 250–180, 180–125, 125–75, 75–38, and <38 μm.

DSC recoded from 20 to 220 °C at a heating rate of 10 °C/min.

Cholesterol adsorption of BKBP

As shown in Fig. 4A, the adsorption capacity for cholesterol significantly increased with the reduction of particle size. The BKBP with a size of <38 μm showed the strongest adsorption capacity (27.27 mg/g); thus, it was chosen for further evaluation. The cholesterol adsorption capacity decreased dramatically from 26.95 to 15.51 mg/g with adsorbent dosage increasing (Fig. 4B). With the increase in initial concentration of cholesterol, the adsorption capacity increased (Fig. 4C). Furthermore, the cholesterol adsorption capacity for different adsorption time (min) and temperature (°C) was shown in Figs. 4D and 4E, respectively. The adsorption increased quickly with increasing time from 10 to 60 min, reaching a plateau in the following 60–150 min (Fig. 4D); nevertheless, the cholesterol adsorption capacity decreased when temperature increased (Fig. 4E).

Figure 4 Cholesterol adsorption capacity of BKBP.

(A) Different particle size, (B) powder dosage, (C) initial cholesterol concentration, (D) absorption time, (E) temperature, and (F) separation factor RL for the Langmuir isotherm.

Adsorption isotherms analysis

The relationship between the adsorption capacity (qe) and the concentration of cholesterol at equilibrium (Ce) was further analyzed via fitting to Langmuir and Freundlich isotherms models, respectively (Ngah & Hanafiah, 2008). The Langmuir model is considered as a monolayer adsorption processing, which assumes monolayer adsorption onto an adsorbent surface. The linear equation is given by 1/qe=[1/(KL×qmax)]/(1/Ce), where qmax represents the maximum adsorption capacity (mg/g), Ce represents the concentration of adsorbate (mg/mL) at equilibrium, and KL represents a constant related to energy of adsorption, which quantitatively reflects the affinity between adsorbent and adsorbate. The maximum adsorption capacity of cholesterol adsorption was calculated as 53.476 mg/g for BKBP (Table 3). Moreover, the essential feature of the Langmuir model was expressed with a dimensionless constant separation factor (RL), which was calculated using the equation of 1/(1 + KL × C0), where C0 represents the initial cholesterol concentration (mg/mL). Therefore, the RL was 0.370–0.804 for the initial cholesterol concentration ranging from 0.25 to 1.50 mg/mL, respectively, indicating a favorable adsorption of cholesterol using the BKBP (Fig. 4F).

Table 3 The results of fitted isothermal adsorption models and their parameters.

Isotherm model	Parameters	Equation	R2	
Langmuir isotherm	qm/(mg/g)	53.476	1qe=0.0192Ce+0.0187	0.911	
KL/(mL/mg)	0.974	
Freundlich isotherm	KF/(mg/g)	0.161	lgqe=0.697lgCe+1.449	0.914	
1/n	0.697	

The Freundlich isotherm model was considered to be multilayer adsorption and could be suitable to highly heterogeneous surface, which could be expressed with the linear equation of lg qe = Ce/n + lg KF, where KF and n represent the Freundlich constants indicative of the adsorption capacity and intensity, respectively. The value of 1/n determined via the Freundlich isotherm was 0.697 (1/n <1) (Table 3), confirming the high adsorption efficient of BKBP.

Antioxidant activity analysis

The antioxidant activity of the BKBP with different particle sizes was evaluated by different in vitro assays (Fig. 5). Regarding radical scavenging activity using DPPH assay, the finer powders with particle sizes of 75–38 and <38 μm exhibited higher DPPH scavenging activities of 87.30% ± 1.77% and 89.40% ± 0.81%, respectively (Fig. 5A). As shown in Fig. 5B, the powder with particle size of 75–38 μm exhibited the strongest hydroxyl radical scavenging activity (88.92% ± 1.38%) among all tested samples, while the BKBP with the particle size of 250–180 μm, obtained via coarse grinding, showed the lowest activity (69.72% ± 2.49%) (Fig. 5B). Furthermore, an increase in ferrous ion-chelating effects was observed when the particle size decreased from 250–180 to 75–38 μm, and the BKBP with a size of 75–38 μm exhibited the strongest chelating activity of 81.16% ± 1.72% (Fig. 5C).

Figure 5 Antioxidant properties of BKBP with particle sizes of 250–180, 180–125, 125–75, 75–38, and <38 μm.

(A) DPPH scavenging activity (%), (B) hydroxyl radical-scavenging activity (%), and (C) ferrous ion-chelating activity (%).

Discussion

Taking into account the nutritional and economical aspects of black kidney beans, fortifying varied bean powders appears to be promising for the production of health food (Lee, Hung & Chou, 2008). Conventional milling methods have generally been used in the pulverization and research of kidney bean (Anton, Fulcher & Arntfield, 2009; Malav et al., 2016). However, until now, systematic studies on superfine kidney bean powder are still limited.

In the present study, BKBP with sizes between 180 and <38 μm was prepared via eccentric vibratory mill. Elliptical, circular and linear vibrations could be generated via eccentric vibratory mill instead of homogeneous circular vibrations, which would increase the amplitude of the individual grinding media and increase the rotational speed of the grinding media filling (Gock & Kurrer, 1999). Consequently, as shown in the particle size and SEM analyses (Table 1; Fig. 1), BKBPs were efficiently broken into smaller fractions, and the shape and original structure of particles were changed to be smoother by the inhomogenous impact force. Therefore, physical–chemical properties of BKBP would be altered with the sieving of special size parameters (250–180, 180–125, 125–75, 75–38, and <38 μm), confirmed the importance of micronization equipment on the fluidity, dissolution, and surface activity of powders (Muttakin, Kim & Lee, 2015).

The color parameters of BKBP (Table 2) depicted their relations to particle size and morphology. The increase of L* values was as expected with the reduction of particle size, due to the increase in surface area, and that would allow more reflection of light (Ahmed et al., 2016). Meanwhile, the loss of pigment and the exposure of internal materials during superfine grinding also could contribute to the improvement of brightness. Thus, the BKBP with a size of <38 μm was brighter, which might be favorable for the applications as food ingredients. The decrease of bulk density and increase of tapped density of fine powders (Table 2) exhibited the enhancement of inter-particulate interactions, which indicated the improvement of flowability of the BKBP. Moreover, according to Table 2, the decreasing angle of repose and slide of superfine BKBP with smaller size also might indicate the increase of flowability (Zhao et al., 2010). But the result was not in agreement with the investigation of Lee & Yoon (2015), who found that the soybean powders with the smallest particle size (250–150 μm) showed a poor flowability because of the cohesion. However, Fu et al. (2012) reported that powder shape significantly affected the flow characteristics of the powder, and stated that more circular and smooth shaped particles had the higher flowability, which was consistent with our results for the morphology analysis (Fig. 1), and confirmed the efficiency of eccentric vibratory mill. Thus, the BKBP with a size of <38 μm had a larger number of particles per unit weight and achieved the higher flowability, which would be beneficial to fill tablets or capsule products to achieve homogeneity state when mixed with other additives.

Furthermore, the decrease of particle size has a substantial effect on the hydration properties of BKBP. Particles with a size of <38 μm exhibited the highest hydratability during soaking (Fig. 2), which was higher than previous data on soybean flours (4.1 g/g) (Heywood et al., 2002) and superfine wheat (Triticum aestivm L.) bran superfine powder (7.0 g/g) (He et al., 2018). Superfine grinding treatment might result in the surface properties changes of the BKBP, such as the increase of surface energy, greater surface area, and the exposure of hydrophilic groups, which led to an easy integration with water (Zhao et al., 2009a). Additionally, high content of protein (20–30%) within the BKBP could also held water through weak forces, such as hydrogen bonds (Shi et al., 2016). Similar results were also confirmed by Chen et al. (2015a) and Zhao et al. (2009a, 2010). In contrast, Raghavendra et al. (2006) found that the hydration properties of coconut dietary fiber were decreased when its particle size was decreased from 550 to 390 μm. It has been reported that grinding the dry fibrous material to fine powder adversely affected its WHC and swelling capacity, presumably attributed to the collapse of the fiber matrix by milling (Kethireddipalli et al., 2002). Hence, various physicochemical characteristic would be discovered because of the diversity of materials and grinding treatments. Eccentric vibratory milling treatment might result in the damage on BKBP structure, and the particle size would be too smaller to compensate differences on the hydration properties. High hydration capacities of BKBP would increase the affinity between the powder and water, and might keep more water in the inner part (He et al., 2018), which would lead to the enhancement of evaporation energy, and exhibited an improved thermal stability (Fig. 3). Therefore, BKBP with the smaller size (such as <38 μm) could be more suitable for water retention, and might thus be more potentially applied as functional ingredient to prevent syneresis and improve textural properties, as well as be utilized in the higher-temperature processing, such as baking or steaming.

It is well known that the surplus cholesterol in the human body forms an initial pathogenic factor of arteriosclerosis, resulting in apoplectic stroke, angina pectoris, and cardio sclerosis (Soh, Kim & Lee, 2003). Food material as biosorbent for cholesterol reducers/extractors is of growing interest, due to many advantages, such as natural, wide availability, healthy, and nontoxicity. Good cholesterol binding capacities have been found using the cereal brans, such as rice bran, oat bran, wheat bran, and corn bran (Kahlon & Chow, 2000). Adsorption properties of four legume seeds (green lentil, white small bean, yellow pea, and yellow soybean) have been evaluated by Górecka, Korczak & Flaczyk (2003), grinding degree was found to be significantly influenced the adsorption properties. In the present study, superfine BKBP was found to have a high cholesterol adsorption capacity by in vitro assays, which was probably correlated with their high contents of dietary fiber, especially hemicelluloses and lignin (Górecka, Korczak & Flaczyk, 2003), thus it could be recommended in the in lipid disorders prophylactic. Particle size, powder dosage, initial concentration of cholesterol, absorption time, and absorption temperature were all found to significantly affect the cholesterol adsorption of BKBP (Fig. 4). Decreasing particle size could effectively improve cholesterol adsorption capacity of BKBP due to the increase of specific surface area, thus lead to a larger contact area with cholesterol and shorter absorbing path distance (Chen et al., 2015b). It was interesting that the relative lower temperature would be favorable for the cholesterol adsorption, thus cholesterol adsorption process using BKBP should be controlled at below 18 °C for a 60 min reaction.

Furthermore, the maximal adsorption capacity (53.476 mg/g) of BKBP was successfully predicted by Langmuir adsorption isotherms analysis (Table 3), which was higher than the ability of okra superfine powder around 18.75 mg/g (Chen et al., 2015b), and carrot pomace insoluble dietary fiber around 30 mg/g (Yu et al., 2018), but lower than thyme (Thymus vulgaris L.) powder of 84.74 mg/g (Salehi et al., 2018). Besides, the value of 1/n (0.697) obtained from the Freundlich model was less than unity, indicating the favorability of the adsorption. The two fitting models suggested that BKBP would be effective as a potential adsorbent. Therefore, it seemed that the BKBP could be applied in the functional food manufactures, such as biscuits and other healthy products, to reduce calories and cholesterol without loss in physical and structural properties (Prokopov, 2014). It has been confirmed that the plant source of seed powder might have hypolipidemic effect on diabetic patients (Kassaian et al., 2009). Thus, superfine BKBP might act as a novel nutraceutical additive/excipient in tablets, such as simvastatin, to provide synergistic effects for lowering serum cholesterol level (Swami et al., 2010). Besides, it would be also interesting to employ the BKBP as the potential biosorption materials in the developments of low cholesterol milk or milk beverages (Oliveira et al., 2015), and even in the extracorporeal perfusion to immediately reduce the content of the lipids in the blood (Salehi et al., 2018).

In addition, multiple antioxidant assays including DPPH and hydroxyl radical scavenging activity, as well as ferrous ion-chelating effects, were carried out in the experiments, and particle sizes showed significant effects on the activities (Fig. 5). The capability of stable free radical 2,2-diphenyl-1-picrylhydrazyl to react with H-donors, including phenolics in natural materials, could be evaluated by the DPPH• test in the visible region after a fixed incubation time (Roginsky & Lissi, 2005; López-Alarcón & Denicola, 2013). In this study, higher DPPH• scavenging activity was obtained with the decrease of BKBP particle size (Fig. 5A). Meanwhile, the powder with a particle size of 75–38 μm exhibited the strongest hydroxyl radical scavenging activity in Fig. 5B. Hydroxyl radicals (•OH) are the most commonly formed reactive oxygen species and have been linked to many clinical disorders, such as brain ischemia, cardiovascular disease, and carcinogenesis (Hu, Chen & Ni, 2012). Several reports also indicated that the •OH scavenging effects could be related to hypoglycemic activity (Chen et al., 2009; Chen, Zhang & Xie, 2005). As shown in Fig. 5C, the ferrous ion (Fe2+)-chelating activity of the BKBP was favorably affected by the reduction of particle size, which would prevent the generation of free radicals, oxyradicals, and lipid peroxidation (Singh & Rajini, 2004).

According to the previous studies, polyphenols and flavonoids compounds were main antioxidant compounds presenting in kidney bean (P. vulgaris L.), containing free and bound forms (Cardador-Martínez, Loarca-Piña & Oomah, 2002; Malav et al., 2016). Meanwhile, as one kind of black coat bean, a high accumulation of anthocyanins relating to antioxidant activity would be found in the epidermis palisade layer, and was up to 13,955 (mg CGE/kg) (Žilić et al., 2013). Therefore, the increase of antioxidant availability in the BKBP with the smaller particle sizes (such as 75–38 and <38 μm) might be attributed to the fact that finer particles would be beneficial for the dissolution of free-form antioxidant compounds. In addition, superfine grinding broke the structure of protein and fiber matrix (as shown in SEM images), and thus increased the availability of bound-form antioxidant compounds linked or embedded in the matrix. However, as shown in Figs. 5B and 5C, compared with the sample size of 75–38 μm, the antioxidant activities of BKBP with a size of <38 μm have a slight decrease (P > 0.05), which might be attributed to the inevitable mechanical impact and heating effect during superfine grinding, leading to altering or disrupting of antioxidant compounds within BKBP. Therefore, controlling grinding degree is of importance, as it will influence powders’ functional properties, and superfine BKBP with a size of 75–38 μm exhibited a potential application as antioxidative products.

Conclusions

Fine BKBP with smooth surface was obtained using the eccentric vibratory milling, and the application potential of BKBP was improved with the decrease of particle size. The BKBP with a particle size of <38 μm exhibited good flowability, hydration properties, and thermal stability. Adsorption isotherm analysis highlighted the promising potential of the superfine BKBP with a particle size of <38 μm as a cholesterol sorbent or an alternative source to absorb harmful lipids. Moreover, compared with the other particle sizes, the superfine BKBP with sizes of <75 μm showed improved antioxidant activities in the free radical scavenging activities. Overall, the BKBP prepared by eccentric vibratory mill with a particle size of <38 μm showed great potentials in the food industry and pharmaceutical field for the health product developments. In the future, in vivo evaluations of the BKBP would be urgently carried out, and the BKBP produced using the eccentric vibratory mill should be further evaluated under various processing conditions to better understand the attributes of the grinding technology.

Supplemental Information

Supplemental Information 1 Raw data hydration properties of BKBP.

Hydration properties of BKBP with particle sizes was determined as described in the Methods section and the data is presented in Figure 2.

Click here for additional data file.

Supplemental Information 2 Raw data DSC curves of BKBP.

DSC curves of BKBP were determined as described in the Methods section and final data is presented in Figure 3.

Click here for additional data file.

Supplemental Information 3 Raw data cholesterol adsorption capacity and adsorption isotherms analysis.

Cholesterol adsorption capacity and adsorption isotherms analysis were determined as described in the Methods section. Data is shown in Figure 4 and Table 3.

Click here for additional data file.

Supplemental Information 4 Raw data antioxidant properties of BKBP.

Antioxidant properties of BKBP presented in Figure 5 were determined by DPPH and hydroxyl radical scavenging activity, as well as ferrous ion-chelating effects.

Click here for additional data file.

We thank Prof. Zhaojun Wei, School of Food and Biological Engineering, Hefei University of Technology, for expertise and help during this research, and members in our laboratory for fruitful discussions. The authors are grateful to the anonymous reviewers’ careful works and thoughtful suggestions.

Additional Information and Declarations

Competing Interests

Author Contributions

Data Availability

The authors declare that they have no competing interests.

Xianbao Sun conceived and designed the experiments, performed the experiments, analyzed the data, prepared figures and/or tables, authored or reviewed drafts of the paper, approved the final draft.

Yuwei Zhang conceived and designed the experiments, performed the experiments, prepared figures and/or tables, approved the final draft.

Jing Li analyzed the data, prepared figures and/or tables, authored or reviewed drafts of the paper, approved the final draft.

Nayab Aslam authored or reviewed drafts of the paper, approved the final draft.

Hanju Sun analyzed the data, contributed reagents/materials/analysis tools, approved the final draft.

Jinlong Zhao performed the experiments, analyzed the data, prepared figures and/or tables, approved the final draft.

Zeyu Wu analyzed the data, contributed reagents/materials/analysis tools, approved the final draft.

Shudong He conceived and designed the experiments, analyzed the data, contributed reagents/materials/analysis tools, authored or reviewed drafts of the paper, approved the final draft.

The following information was supplied regarding data availability:

The raw data are available in the Supplemental Files.

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
