# Peer review of "Effects of particle size on physicochemical and functional properties of superfine black kidney bean (Phaseolus vulgaris L.) powder"

_PeerJ, doi:10.7717/peerj.6369_

## Round 0.1 · original submission · Major Revisions

To bring your manuscript to a state of acceptability you must respond to each of the reviewer's comments and either alter your manuscript as requested or state why you consider the suggestion inappropriate.

Reviewer 1 ·

Basic reporting

These aspects of the manuscript are fine.

Experimental design

All of these are fine, but regarding the second point (research question well defined, relevant & meaningful, stated how research fills an identified knowledge gap), see the general comments.

Validity of the findings

see general comments

Additional comments

The manuscript is well written and easy to understand, and the data are presented in a clear manner using typical formats. However it is not clear what new information is presented in the study, in light of the background literature discussed in lines 48-71 of the introduction. It is well known that grinding any food material to a very fine powder will make it easier to extract components from it. Also, it is obvious that finely ground foodstuffs will absorb water faster and better than coarser materials. So it needs to be explained how super-finely ground black kidney bean flours are somehow unexpectedly different than anything else so treated. The physical data including the SEM images serve only to confirm that the material has been reduced in size by the grinding, with the inevitable changes in surface area, flow properties, etc. The data that show higher amount of antioxidants in superfine flour are misleading, since it is obvious that more antioxidants are extractable and thus detectable in the finer flour. Antioxidants are not synthesized or produced by grinding, nor can they be increased by mechanical degradation of other components by grinding. It cannot be concluded that superfine flour has more of something the detection of which is a function of the particle size.

The use of superfine black kidney bean flour to absorb cholesterol from animal fats is hard to imagine as a practical approach. If such a use is proposed, at least it would be necessary to compare the performance of the flour with other adsorbents such as activated charcoal or clay, for example. Foods prepared with superfine black kidney bean flours would not absorb cholesterol from other foods consumed in the diet, as the flour components would be incorporated into the food and thus not be in a superfine state after ingestion.

Some aspect of the superfine black kidney bean flour that is different from other superfine kidney bean flours, other superfine pulse flours, other superfine grain flours, or other superfine foods in general needs to be shown, other than properties of the black kidney beans themselves, which are known.

Reviewer 2 ·

Basic reporting

The manuscript is fairly well-written, well-organised and easy to read.

Experimental design

In the manuscript, the procedures for the experimental techniques were adequately described although in some aspects, improvements can be made by providing the necessary details.

1. It is common to have a botanist to authenticate the samples that are used in any study.
2. Please state the time used to mill the beans in the domestic disc-mill. Would be beans be overheated during the 10 min milling time?
3. How did the authors choose a concentration of 5 mg/ml for two of the antioxidant assays? The concentration of samples used for the hydroxyl radical scavenging assay was not stated. The metal chelating assay is not referenced.
4. Did the authors suspend the milled bean samples into water and directly use them for the antioxidant assays? Why this method is chosen instead of preparing extracts from the milled samples?

Validity of the findings

Overall, the authors presented their results in a systematic manner and interpreted their data carefully.

Additional comments

The manuscript described the effects of particle size of black kidney beans on their physicochemical and functional properties. The authors characterised the milled samples in terms of particle size distribution, surface area size, colour, flow property, water holding and water retention capacity, as well as thermal property.This part is fairly straight forward. In addition, the bean samples were also demonstrated to have cholesterol absorption and antioxidant properties. Overall, I think the manuscript contains findings that might be useful for the nutraceutical industry. Nonetheless, several issues need to be addresed:

The authors need to justify why dose-response studies have not been carried out (in place of a single dose analysis, and how the dose was chosen). It is a standard practice to express the results in terms of IC50 values so that comparisons can be made with appropriate positive controls (missing data) and data in the literature. The authors stated that their findings on the correlation of particle size of samples and increase in antioxidant activity were in agreement with previous studies but there is little attempt to discuss the possible reasons, which in turn needs to be supported by evidence from the literature. The manuscript will also benefit from a more thorough discussion, perhaps the authors can compare and contrast their results with previous related work. A mention of some potential further work is suggested.

---

## Round 0.2 · Minor Revisions

The editor agrees to the Reviewer 1's comment regarding few issues related to the clarity of the discussion section. The authors are given once more chance to address this major deficiency.

Reviewer 1 ·

Basic reporting

no comment

Experimental design

no comment

Validity of the findings

The rationale and benefit to the literature is not clearly stated, as most if not all of the results have previously been cited or are obvious consequences of the treatment.

Additional comments

The responses and revisions based my comments, though lengthy, are tangential and circumlocutory, and do not really address the points made. In response to the comment about data showing that the particles become smaller and have different shapes after superfine grinding, more data and references that confirm that obvious fact are now presented. In response to the comment that finer samples are more extractable and hydratable and therefore could be expected to give different analysis results, again it is repeated with more literature precedents and explanations, some contradictory. In response to the comment that using superfine red kidney bean flour as a cholesterol absorbent is an arbitrary and expensive option, some additional references were added and an example of using it as filler in simvastatin tablets was proposed. Recycled newsprint contains the same components stated to be relevant to this functionality in SRKBF. It is still not clear what is unexpected about any of the experimental findings, even though the manuscript is now somewhat longer. If possible, the authors should consider addressing some of these issues in the discussion.

---

## Round 0.3 · Minor Revisions

Please attend to the additional comment made by Reviewer #1.

Reviewer 1 ·

Basic reporting

no comment

Experimental design

no comment

Validity of the findings

The editor will have to decide whether this manuscript is a repetition of well-known, widely accepted results.

Additional comments

Please add an explanation of why it is unexpected that finer particle size will increase water uptake efficiency, make plant constituents more extractable, or absorb more cholesterol, and why black kidney beans would exhibit different responses to this treatment than any other pulse or grain product.

Reviewer 2 ·

Basic reporting

No comment

Experimental design

No comment

Validity of the findings

No comment

Additional comments

The manuscript has been revised based on the comments of the reviewer. I agree to some of the previous reviewer's comment and the authors have attempted to address them. Nonetheless, I do think the findings presented here have merits to be considered for publication. Language correction is recommended to improve clarity in some parts of the manuscript.

---

## Round 0.4 · accepted · Accept

Thank you for taking your time and effort in improving the manuscript.

#